# Convergence Rates of Active Learning for Maximum Likelihood Estimation

Kamalika Chaudhuri [*]     Sham M. Kakade [†]     Praneeth Netrapalli [‡]     Sujay Sanghavi [§]

## Abstract

An active learner is given a class of models, a large set of unlabeled examples, and the ability to interactively query labels of a subset of these examples; the goal of the learner is to learn a model in the class that fits the data well.

Previous theoretical work has rigorously characterized label complexity of active learning, but most of this work has focused on the PAC or the agnostic PAC model. In this paper, we shift our attention to a more general setting – maximum likelihood estimation. Provided certain conditions hold on the model class, we provide a two-stage active learning algorithm for this problem. The conditions we require are fairly general, and cover the widely popular class of Generalized Linear Models, which in turn, include models for binary and multi-class classification, regression, and conditional random fields.

We provide an upper bound on the label requirement of our algorithm, and a lower bound that matches it up to lower order terms. Our analysis shows that unlike binary classification in the realizable case, just a single extra round of interaction is sufficient to achieve near-optimal performance in maximum likelihood estimation. On the empirical side, the recent work in [12] and [13] (on active linear and logistic regression) shows the promise of this approach.

## 1   Introduction

In active learning, we are given a sample space $\mathcal{X}$, a label space $\mathcal{Y}$, a class of models that map $\mathcal{X}$ to $\mathcal{Y}$, and a large set $U$ of unlabelled samples. The goal of the learner is to learn a model in the class with small target error while interactively querying the labels of as few of the unlabelled samples as possible.

Most theoretical work on active learning has focussed on the PAC or the agnostic PAC model, where the goal is to learn *binary classifiers* that belong to a particular hypothesis class [2, 14, 10, 7, 3, 4, 22], and there has been only a handful of exceptions [19, 9, 20]. In this paper, we shift our attention to a more general setting – maximum likelihood estimation (MLE), where $\Pr(Y|X)$ is described by a model $\theta$ belonging to a model class $\Theta$. We show that when data is generated by a model in this class, we can do active learning provided the model class $\Theta$ has the following simple property: the Fisher information matrix for any model $\theta \in \Theta$ at any $(x, y)$ depends only on $x$ and $\theta$. This condition is satisfied in a number of widely applicable model classes, such as Linear Regression and Generalized Linear Models (GLMs), which in turn includes models for Multiclass Classification and Conditional Random Fields. Consequently, we can provide active learning algorithms for maximum likelihood estimation in all these model classes.

The standard solution to active MLE estimation in the statistics literature is to select samples for label query by optimizing a class of summary statistics of the asymptotic covariance matrix of the

---

[*]Dept. of CS, University of California at San Diego. Email: `kamalika@cs.ucsd.edu`

[†]Dept. of CS and of Statistics, University of Washington. Email: `sham@cs.washington.edu`

[‡]Microsoft Research New England. Email:`praneeth@microsoft.com`

[§]Dept. of ECE, The University of Texas at Austin. Email:`sanghavi@mail.utexas.edu`

estimator [6]. The literature, however, does not provide any guidance towards which summary statistic should be used, or any analysis of the solution quality when a finite number of labels or samples are available. There has also been some recent work in the machine learning community [12, 13, 19] on this problem; but these works focus on simple special cases (such as linear regression [19, 12] or logistic regression [13]), and only [19] involves a consistency and finite sample analysis.

In this work, we consider the problem in its full generality, with the goal of minimizing the expected log-likelihood error over the unlabelled data. We provide a two-stage active learning algorithm for this problem. In the first stage, our algorithm queries the labels of a small number of random samples from the data distribution in order to construct a crude estimate $\theta_1$ of the optimal parameter $\theta^*$. In the second stage, we select a set of samples for label query by optimizing a summary statistic of the covariance matrix of the estimator at $\theta_1$; however, unlike the experimental design work, our choice of statistic is directly motivated by our goal of minimizing the expected log-likelihood error, which guides us towards the right objective.

We provide a finite sample analysis of our algorithm when some regularity conditions hold and when the negative log likelihood function is convex. Our analysis is still fairly general, and applies to Generalized Linear Models, for example. We match our upper bound with a corresponding lower bound, which shows that the convergence rate of our algorithm is optimal (except for lower order terms); the finite sample convergence rate of any algorithm that uses (perhaps multiple rounds of) sample selection and maximum likelihood estimation is either the same or higher than that of our algorithm. This implies that unlike what is observed in learning binary classifiers, a single round of interaction is sufficient to achieve near-optimal log likelihood error for ML estimation.

## 1.1 Related Work

Previous theoretical work on active learning has focussed on learning a classifier belonging to a hypothesis class $\mathcal{H}$ in the PAC model. Both the realizable and non-realizable cases have been considered. In the realizable case, a line of work [7, 18] has looked at a generalization of binary search; while their algorithms enjoy low label complexity, this style of algorithms is inconsistent in the presence of noise. The two main styles of algorithms for the non-realizable case are disagreement-based active learning [2, 10, 4], and margin or confidence-based active learning [3, 22]. While active learning in the realizable case has been shown to achieve an exponential improvement in label complexity over passive learning [2, 7, 14], in the agnostic case, the gains are more modest (sometimes a constant factor) [14, 10, 8]. Moreover, lower bounds [15] show that the label requirement of any agnostic active learning algorithm is always at least $\Omega(\nu^2/\epsilon^2)$, where $\nu$ is the error of the best hypothesis in the class, and $\epsilon$ is the target error. In contrast, our setting is much more general than binary classification, and includes regression, multi-class classification and certain kinds of conditional random fields that are not covered by previous work.

[19] provides an active learning algorithm for linear regression problem under model mismatch. Their algorithm attempts to learn the location of the mismatch by fitting increasingly refined partitions of the domain, and then uses this information to reweight the examples. If the partition is highly refined, then the computational complexity of the resulting algorithm may be exponential in the dimension of the data domain. In contrast, our algorithm applies to a more general setting, and while we do not address model mismatch, our algorithm has polynomial time complexity. [1] provides an active learning algorithm for Generalized Linear Models in an online selective sampling setting; however, unlike ours, their input is a stream of unlabelled examples, and at each step, they need to decide whether the label of the current example should be queried.

Our work is also related to the classical statistical work on optimal experiment design, which mostly considers maximum likelihood estimation [6]. For uni-variate estimation, they suggest selecting samples to maximize the Fisher information which corresponds to minimizing the variance of the regression coefficient. When $\theta$ is multi-variate, the Fisher information is a matrix; in this case, there are multiple notions of *optimal design* which correspond to maximizing different parameters of the Fisher information matrix. For example, D-optimality maximizes the determinant, and A-optimality maximizes the trace of the Fisher information. In contrast with this work, we directly optimize the expected log-likelihood over the unlabelled data which guides us to the appropriate objective function; moreover, we provide consistency and finite sample guarantees.

Finally, on the empirical side, [13] and [12] derive algorithms similar to ours for logistic and linear regression based on projected gradient descent. Notably, these works provide promising empirical evidence for this approach to active learning; however, no consistency guarantees or convergence rates are provided (the rates presented in these works are not stated in terms of the sample size). In contrast, our algorithm applies more generally, and we provide consistency guarantees and convergence rates. Moreover, unlike [13], our logistic regression algorithm uses a single extra round of interaction, and our results illustrate that a single round is sufficient to achieve a convergence rate that is optimal except for lower order terms.

## 2   The Model

We begin with some notation. We are given a pool $U = \{x_1, \ldots, x_n\}$ of $n$ unlabelled examples drawn from some instance space $\mathcal{X}$, and the ability to interactively query labels belonging to a label space $\mathcal{Y}$ of $m$ of these examples. In addition, we are given a family of models $\mathcal{M} = \{p(y|x, \theta), \theta \in \Theta\}$ parameterized by $\theta \in \Theta \subseteq \mathbb{R}^d$. We assume that there exists an unknown parameter $\theta^* \in \Theta$ such that querying the label of an $x_i \in U$ generates a $y_i$ drawn from the distribution $p(y|x_i, \theta^*)$. We also abuse notation and use $U$ to denote the uniform distribution over the examples in $U$.

We consider the *fixed-design* (or *transductive*) setting, where our goal is to minimize the error on the fixed set of points $U$. For any $x \in \mathcal{X}, y \in \mathcal{Y}$ and $\theta \in \Theta$, we define the negative log-likelihood function $L(y|x, \theta)$ as:
$$L(y|x, \theta) = -\log p(y|x, \theta)$$
Our goal is to find a $\hat{\theta}$ to minimize $L_U(\hat{\theta})$, where
$$L_U(\theta) = \mathbb{E}_{X \sim U, Y \sim p(Y|X, \theta^*)}[L(Y|X, \theta)]$$
by interactively querying labels for a subset of $U$ of size $m$, where we allow label queries with replacement i.e., the label of an example may be queried multiple times.

An additional quantity of interest to us is the Fisher information matrix, or the Hessian of the negative log-likelihood $L(y|x, \theta)$ function, which determines the convergence rate. For our active learning procedure to work correctly, we require the following condition.

**Condition 1.** *For any $x \in \mathcal{X}$, $y \in \mathcal{Y}$, $\theta \in \Theta$, the Fisher information $\frac{\partial^2 L(y|x, \theta)}{\partial \theta^2}$ is a function of only $x$ and $\theta$ (and does not depend on $y$.)*

Condition 1 is satisfied by a number of models of practical interest; examples include linear regression and generalized linear models. Section 5.1 provides a brief derivation of Condition 1 for generalized linear models.

For any $x$, $y$ and $\theta$, we use $I(x, \theta)$ to denote the Hessian $\frac{\partial^2 L(y|x, \theta)}{\partial \theta^2}$; observe that by Assumption 1, this is just a function of $x$ and $\theta$. Let $\Gamma$ be any distribution over the unlabelled samples in $U$; for any $\theta \in \Theta$, we use:
$$I_\Gamma(\theta) = \mathbb{E}_{X \sim \Gamma}[I(X, \theta)]$$

## 3   Algorithm

The main idea behind our algorithm is to sample $x_i$ from a well-designed distribution $\Gamma$ over $U$, query the labels of these samples and perform ML estimation over them. To ensure good performance, $\Gamma$ should be chosen carefully, and our choice of $\Gamma$ is motivated by Lemma 1. Suppose the labels $y_i$ are generated according to: $y_i \sim p(y|x_i, \theta^*)$. Lemma 1 states that the expected log-likelihood error of the ML estimate with respect to $m$ samples from $\Gamma$ in this case is essentially $\text{Tr}\left(I_\Gamma(\theta^*)^{-1} I_U(\theta^*)\right)/m$.

This suggests selecting $\Gamma$ as the distribution $\Gamma^*$ that minimizes $\text{Tr}\left(I_{\Gamma^*}(\theta^*)^{-1} I_U(\theta^*)\right)$. Unfortunately, we cannot do this as $\theta^*$ is unknown. We resolve this problem through a two stage algorithm; in the first stage, we use a small number $m_1$ of samples to construct a coarse estimate $\theta_1$ of $\theta^*$ (Steps 1-2). In the second stage, we calculate a distribution $\Gamma_1$ which minimizes $\text{Tr}\left(I_{\Gamma_1}(\theta_1)^{-1} I_U(\theta_1)\right)$ and draw samples from (a slight modification of) this distribution for a finer estimation of $\theta^*$ (Steps 3-5).

---

**Algorithm 1** ActiveSetSelect

---

**Input:** Samples $x_i$, for $i = 1, \cdots, n$

1: Draw $m_1$ samples u.a.r from $U$, and query their labels to get $S_1$.
2: Use $S_1$ to solve the MLE problem:

$$\theta_1 = \text{argmin}_{\theta \in \Theta} \sum_{(x_i, y_i) \in S_1} L(y_i | x_i, \theta)$$

3: Solve the following SDP (refer Lemma 3):

$$a^* = \text{argmin}_a \, \text{Tr}\left(S^{-1} I_U(\theta_1)\right) \quad \text{s.t.} \quad \left\{ \begin{array}{c} S = \sum_i a_i I(x_i, \theta_1) \\ 0 \leq a_i \leq 1 \\ \sum_i a_i = m_2 \end{array} \right.$$

4: Draw $m_2$ examples using probability $\overline{\Gamma} = \alpha \Gamma_1 + (1 - \alpha)U$ where the distribution $\Gamma_1 = \frac{a_i^*}{m_2}$ and $\alpha = 1 - m_2^{-1/6}$. Query their labels to get $S_2$.
5: Use $S_2$ to solve the MLE problem:

$$\theta_2 = \text{argmin}_{\theta \in \Theta} \sum_{(x_i, y_i) \in S_2} L(y_i | x_i, \theta)$$

**Output:** $\theta_2$

---

The distribution $\Gamma_1$ is modified slightly to $\overline{\Gamma}$ (in Step 4) to ensure that $I_{\overline{\Gamma}}(\theta^*)$ is well conditioned with respect to $I_U(\theta^*)$.

The algorithm is formally presented in Algorithm 1.

Finally, note that Steps 1-2 are necessary because $I_U$ and $I_\Gamma$ are functions of $\theta$. In certain special cases such as linear regression, $I_U$ and $I_\Gamma$ are independent of $\theta$. In those cases, Steps 1-2 are unnecessary, and we may skip directly to Step 3.

## 4  Performance Guarantees

The following regularity conditions are essentially a quantified version of the standard Local Asymptotic Normality (LAN) conditions for studying maximum likelihood estimation (see [5, 21]).

**Assumption 1.** *(Regularity conditions for LAN)*

1. **Smoothness**: *The first three derivatives of $L(y|x, \theta)$ exist in all interior points of $\Theta \subseteq \mathbb{R}^d$.*

2. **Compactness**: *$\Theta$ is compact and $\theta^*$ is an interior point of $\Theta$.*

3. **Strong Convexity**: *$I_U(\theta^*) = \frac{1}{n} \sum_{i=1}^n I(x_i, \theta^*)$ is positive definite with smallest singular value $\sigma_{min} > 0$.*

4. **Lipschitz continuity**: *There exists a neighborhood $B$ of $\theta^*$ and a constant $L_3$ such that for all $x \in U$, $I(x, \theta)$ is $L_3$-Lipschitz in this neighborhood.*

$$\left\| I_U(\theta^*)^{-1/2} \left(I(x, \theta) - I(x, \theta')\right) I_U(\theta^*)^{-1/2} \right\|_2 \leq L_3 \left\| \theta - \theta' \right\|_{I_U(\theta^*)},$$

   *for every $\theta, \theta' \in B$.*

5. **Concentration at $\theta^*$**: *For any $x \in U$ and $y$, we have (with probability one),*

$$\left\| \nabla L(y|x, \theta^*) \right\|_{I_U(\theta^*)^{-1}} \leq L_1, \text{ and } \left\| I_U(\theta^*)^{-1/2} I(x, \theta^*) I_U(\theta^*)^{-1/2} \right\|_2 \leq L_2.$$

6. **Boundedness**: *$\max_{(x,y)} \sup_{\theta \in \Theta} |L(x, y|\theta)| \leq R$.*

In addition to the above, we need one extra condition which is essentially a pointwise self concordance. This condition is satisfied by a vast class of models, including the generalized linear models.

**Assumption 2.** *Point-wise self concordance:*

$$-L_4 \|\theta - \theta^*\|_2 I(x, \theta^*) \preceq I(x, \theta) - I(x, \theta^*) \preceq L_4 \|\theta - \theta^*\|_2 I(x, \theta^*).$$

**Definition 1.** *[Optimal Sampling Distribution $\Gamma^*$] We define the* optimal sampling distribution $\Gamma^*$ *over the points in $U$ as the distribution $\Gamma^* = (\gamma_1^*, \ldots, \gamma_n^*)$ for which $\gamma_i^* \geq 0$, $\sum_i \gamma_i^* = 1$, and $Tr\left(I_{\Gamma^*}(\theta^*)^{-1} I_U(\theta^*)\right)$ is as small as possible.*

Definition 1 is motivated by Lemma 1, which indicates that under some mild regularity conditions, a ML estimate calculated on samples drawn from $\Gamma^*$ will provide the best convergence rates (including the right constant factor) for the expected log-likelihood error.

We now present the main result of our paper. The proof of the following theorem and all the supporting lemmas will be presented in Appendix A.

**Theorem 1.** *Suppose the regularity conditions in Assumptions 1 and 2 hold. Let $\beta \geq 10$, and the number of samples used in step (1) be $m_1 > \mathcal{O}\left(\max\left(L_2 \log^2 d, L_1^2\left(L_3^2 + \frac{1}{\sigma_{min}}\right)\log^2 d, \frac{diameter(\Theta)}{Tr\left(I_U(\theta^*)^{-1}\right)}, \frac{\beta^2 L_4^2}{\delta} Tr\left(I_U(\theta^*)^{-1}\right)\right)\right)$. Then with probability $\geq 1 - \delta$, the expected log likelihood error of the estimate $\theta_2$ of Algorithm 1 is bounded as:*

$$\mathbb{E}\left[L_U(\theta_2)\right] - L_U(\theta^*) \leq \left(1 + \frac{2}{\beta - 1}\right)^4 (1 + \widetilde{\epsilon}_{m_2}) Tr\left(I_{\Gamma^*}(\theta^*)^{-1} I_U(\theta^*)\right) \frac{1}{m_2} + \frac{R}{m_2^2}, \quad (1)$$

*where $\Gamma^*$ is the optimal sampling distribution in Definition 1 and $\widetilde{\epsilon}_{m_2} = \mathcal{O}\left(\left(L_1 L_3 + \sqrt{L_2}\right)\frac{\sqrt{\log dm_2}}{m_2^{1/6}}\right)$. Moreover, for any sampling distribution $\Gamma$ satisfying $I_\Gamma(\theta^*) \succeq c I_U(\theta^*)$ and label constraint of $m_2$, we have the following lower bound on the expected log likelihood error for ML estimate:*

$$\mathbb{E}\left[L_U(\widehat{\theta}_\Gamma)\right] - L_U(\theta^*) \geq (1 - \epsilon_{m_2}) Tr\left(I_\Gamma(\theta^*)^{-1} I_U(\theta^*)\right) \frac{1}{m_2} - \frac{L_1^2}{cm_2^2}, \quad (2)$$

*where $\epsilon_{m_2} \stackrel{def}{=} \frac{\widetilde{\epsilon}_{m_2}}{c^2 m_2^{1/3}}$.*

**Remark 1.** *(Restricting to Maximum Likelihood Estimation) Our restriction to maximum likelihood estimators is minor, as this is close to minimax optimal (see [16]). Minor improvements with certain kinds of estimators, such as the James-Stein estimator, are possible.*

### 4.1 Discussions

Several remarks about Theorem 1 are in order.

The high probability bound in Theorem 1 is with respect to the samples drawn in $S_1$; provided these samples are representative (which happens with probability $\geq 1 - \delta$), the output $\theta_2$ of Algorithm 1 will satisfy (1). Additionally, Theorem 1 assumes that the labels are sampled *with replacement*; in other words, we can query the label of a point $x_i$ multiple times. Removing this assumption is an avenue for future work.

Second, the highest order term in both (1) and (2) is $\text{Tr}\left(I_{\Gamma^*}(\theta^*)^{-1} I_U(\theta^*)\right)/m$. The terms involving $\epsilon_{m_2}$ and $\widetilde{\epsilon}_{m_2}$ are lower order as both $\epsilon_{m_2}$ and $\widetilde{\epsilon}_{m_2}$ are $o(1)$. Moreover, if $\beta = \omega(1)$, then the term involving $\beta$ in (1) is of a lower order as well. Observe that $\beta$ also measures the tradeoff between $m_1$ and $m_2$, and as long as $\beta = o(\sqrt{m_2})$, $m_1$ is also of a lower order than $m_2$. Thus, provided $\beta$ is $\omega(1)$ and $o(\sqrt{m_2})$, the convergence rate of our algorithm is optimal except for lower order terms.

Finally, the lower bound (2) applies to distributions $\Gamma$ for which $I_\Gamma(\theta^*) \geq c I_U(\theta^*)$, where $c$ occurs in the lower order terms of the bound. This constraint is not very restrictive, and does not affect the asymptotic rate. Observe that $I_U(\theta^*)$ is full rank. If $I_\Gamma(\theta^*)$ is not full rank, then the expected log likelihood error of the ML estimate with respect to $\Gamma$ will not be consistent, and thus such a $\Gamma$ will never achieve the optimal rate. If $I_\Gamma(\theta^*)$ is full rank, then there always exists a $c$ for which $I_\Gamma(\theta^*) \geq c I_U(\theta^*)$. Thus (2) essentially states that for distributions $\Gamma$ where $I_\Gamma(\theta^*)$ is close to being rank-deficient, the asymptotic convergence rate of $O(\text{Tr}\left(I_\Gamma(\theta^*)^{-1} I_U(\theta^*)\right)/m_2)$ is achieved at larger values of $m_2$.

## 4.2 Proof Outline

Our main result relies on the following three steps.

### 4.2.1 Bounding the Log-likelihood Error

First, we characterize the log likelihood error (wrt $U$) of the empirical risk minimizer (ERM) estimate obtained using a sampling distribution $\Gamma$. Concretely, let $\Gamma$ be a distribution on $U$. Let $\widehat{\theta}_\Gamma$ be the ERM estimate using the distribution $\Gamma$:

$$\widehat{\theta}_\Gamma = \operatorname{argmin}_{\theta \in \Theta} \frac{1}{m_2} \sum_{i=1}^{m_2} L(Y_i | X_i, \theta), \tag{3}$$

where $X_i \sim \Gamma$ and $Y_i \sim p(y|X_i, \theta^*)$. The core of our analysis is Lemma 1, which shows a precise estimate of the log likelihood error $\mathbb{E}\left[L_U\left(\widehat{\theta}_\Gamma\right) - L_U\left(\theta^*\right)\right]$.

**Lemma 1.** *Suppose $L$ satisfies the regularity conditions in Assumptions 1 and 2. Let $\Gamma$ be a distribution on $U$ and $\widehat{\theta}_\Gamma$ be the ERM estimate (3) using $m_2$ labeled examples. Suppose further that $I_\Gamma(\theta^*) \succeq c I_U(\theta^*)$ for some constant $c < 1$. Then, for any $p \geq 2$ and $m_2$ large enough such that $\epsilon_{m_2} \stackrel{def}{=} \mathcal{O}\left(\frac{1}{c^2}\left(L_1 L_3 + \sqrt{L_2}\right)\sqrt{\frac{p \log dm_2}{m_2}}\right) < 1$, we have:*

$$(1 - \epsilon_{m_2})\frac{\tau^2}{m_2} - \frac{L_1^2}{cm_2^{p/2}} \leq \mathbb{E}\left[L_U\left(\widehat{\theta}_\Gamma\right) - L_U\left(\theta^*\right)\right] \leq (1 + \epsilon_{m_2})\frac{\tau^2}{m_2} + \frac{R}{m_2^p},$$

*where $\tau^2 \stackrel{def}{=} Tr\left(I_\Gamma(\theta^*)^{-1} I_U(\theta^*)\right)$.*

### 4.2.2 Approximating $\theta^*$

Lemma 1 motivates sampling from the optimal sampling distribution $\Gamma^*$ that minimizes $Tr\left(I_{\Gamma^*}(\theta^*)^{-1} I_U(\theta^*)\right)$. However, this quantity depends on $\theta^*$, which we do not know. To resolve this issue, our algorithm first queries the labels of a small fraction of points ($m_1$) and solves a ML estimation problem to obtain a coarse estimate $\theta_1$ of $\theta^*$.

How close should $\theta_1$ be to $\theta^*$? Our analysis indicates that it is sufficient for $\theta_1$ to be close enough that for any $x$, $I(x, \theta_1)$ is a *constant factor spectral approximation* to $I(x, \theta^*)$; the number of samples needed to achieve this is analyzed in Lemma 2.

**Lemma 2.** *Suppose $L$ satisfies the regularity conditions in Assumptions 1 and 2. If the number of samples used in the first step*

$$m_1 > \mathcal{O}\left(\max\left(L_2 \log^2 d, L_1^2\left(L_3^2 + \frac{1}{\sigma_{min}}\right)\log^2 d, \frac{diameter(\Theta)}{Tr\left(I_U(\theta^*)^{-1}\right)}, \frac{\beta^2 L_4^2}{\delta}Tr\left(I_U(\theta^*)^{-1}\right)\right)\right),$$

*then, we have:*

$$-\frac{1}{\beta} I\left(x, \theta^*\right) \preceq I\left(x, \theta_1\right) - I\left(x, \theta^*\right) \preceq \frac{1}{\beta} I\left(x, \theta^*\right) \ \forall \ x \in X$$

*with probability greater than $1 - \delta$.*

### 4.2.3 Computing $\Gamma_1$

Third, we are left with the task of obtaining a distribution $\Gamma_1$ that minimizes the log likelihood error. We now pose this optimization problem as an SDP.

From Lemmas 1 and 2, it is clear that we should aim to obtain a sampling distribution $\Gamma = \left(\frac{a_i}{m_2} : i \in [n]\right)$ minimizing $Tr\left(I_\Gamma(\theta_1)^{-1} I_U(\theta_1)\right)$. Let $I_U(\theta_1) = \sum_j \sigma_j v_j v_j^\top$ be the singular value decomposition (svd) of $I_U(\theta_1)$. Since $Tr\left(I_\Gamma(\theta_1)^{-1} I_U(\theta_1)\right) = \sum_{j=1}^d \sigma_j v_j^\top I_\Gamma(\theta_1)^{-1} v_j$, this is equivalent

to solving:

$$\min_{a,c} \sum_{j=1}^{d} \sigma_j c_j \quad \text{s.t.} \quad \begin{cases} S = \sum_i a_i I(x_i, \theta_1) \\ v_j^\top S^{-1} v_j \le c_j \\ a_i \in [0,1] \\ \sum_i a_i = m_2. \end{cases} \tag{4}$$

Among the above constraints, the constraint $v_j^\top S^{-1} v_j \le c_j$ seems problematic. However, Schur complement formula tells us that: $\begin{bmatrix} c_j & v_j^\top \\ v_j & S \end{bmatrix} \succeq 0 \iff S \succeq 0$ and $v_j^\top S^{-1} v_j \le c_j$. In our case, we know that $S \succeq 0$, since it is a sum of positive semi definite matrices. The above argument proves the following lemma.

**Lemma 3.** *The following two optimization programs are equivalent:*

$$
\begin{aligned}
&\min_a && Tr\left(S^{-1} I_U(\theta_1)\right) \\
&s.t. && S = \sum_i a_i I(x_i, \theta_1) \\
& && a_i \in [0,1] \\
& && \sum_i a_i = m_2.
\end{aligned}
\quad \equiv \quad
\begin{aligned}
&\min_{a,c} && \sum_{j=1}^{d} \sigma_j c_j \\
&s.t. && S = \sum_i a_i I(x_i, \theta_1) \\
& && \begin{bmatrix} c_j & v_j^\top \\ v_j & S \end{bmatrix} \succeq 0 \\
& && a_i \in [0,1] \\
& && \sum_i a_i = m_2,
\end{aligned}
$$

*where $I_U(\theta_1) = \sum_j \sigma_j v_j v_j^\top$ denotes the svd of $I_U(\theta_1)$.*

## 5 Illustrative Examples

We next present some examples that illustrate Theorem 1. We begin by showing that Condition 1 is satisfied by the popular class of Generalized Linear Models.

### 5.1 Derivations for Generalized Linear Models

A generalized linear model is specified by three parameters – a linear model, a sufficient statistic, and a member of the exponential family. Let $\eta$ be a linear model: $\eta = \theta^\top X$. Then, in a Generalized Linear Model (GLM), $Y$ is drawn from an exponential family distribution with parameter $\eta$. Specifically, $p(Y = y|\eta) = e^{\eta^\top t(y) - A(\eta)}$, where $t(\cdot)$ is the sufficient statistic and $A(\cdot)$ is the log-partition function. From properties of the exponential family, the log-likelihood is written as $\log p(y|\eta) = \eta^\top t(y) - A(\eta)$. If we take $\eta = \theta^\top x$, and take the derivative with respect to $\theta$, we have: $\frac{\partial \log p(y|\theta,x)}{\partial \theta} = x t(y) - x A'(\theta^\top x)$. Taking derivatives again gives us $\frac{\partial^2 \log p(y|\theta,x)}{\partial \theta^2} = -x x^\top A''(\theta^\top x)$, which is independent of $y$.

### 5.2 Specific Examples

We next present three illustrative examples of problems that our algorithm may be applied to.

**Linear Regression.** Our first example is linear regression. In this case, $x \in \mathbb{R}^d$ and $Y \in \mathbb{R}$ are generated according to the distribution: $Y = \theta_*^\top X + \eta$, where $\eta$ is a noise variable drawn from $\mathcal{N}(0,1)$. In this case, the negative loglikelihood function is: $L(y|x,\theta) = (y - \theta^\top x)^2$, and the corresponding Fisher information matrix $I(x,\theta)$ is given as: $I(x,\theta) = x x^\top$. Observe that in this (very special) case, the Fisher information matrix does not depend on $\theta$; as a result we can eliminate the first two steps of the algorithm, and proceed directly to step 3. If $\Sigma = \frac{1}{n} \sum_i x_i x_i^\top$ is the covariance matrix of $U$, then Theorem 1 tells us that we need to query labels from a distribution $\Gamma^*$ with covariance matrix $\Lambda$ such that $\text{Tr}\left(\Lambda^{-1}\Sigma\right)$ is minimized.

We illustrate the advantages of active learning through a simple example. Suppose $U$ is the unlabelled distribution:

$$x_i = \begin{cases} e_1 & \text{w.p. } 1 - \frac{d-1}{d^2}, \\ e_j & \text{w.p. } \frac{1}{d^2} \text{ for } j \in \{2, \cdots, d\}, \end{cases}$$

where $e_j$ is the standard unit vector in the $j^{\text{th}}$ direction. The covariance matrix $\Sigma$ of $U$ is a diagonal matrix with $\Sigma_{11} = 1 - \frac{d-1}{d^2}$ and $\Sigma_{jj} = \frac{1}{d^2}$ for $j \ge 2$. For passive learning over $U$, we query labels

of examples drawn from $U$ which gives us a convergence rate of $\frac{\text{Tr}\left(\Sigma^{-1}\Sigma\right)}{m} = \frac{d}{m}$. On the other hand, active learning chooses to sample examples from the distribution $\Gamma^*$ such that

$$x_i = \begin{cases} e_1 & \text{w.p.} \ \sim \ 1 - \frac{d-1}{2d}, \\ e_j & \text{w.p.} \ \sim \ \frac{1}{2d} \text{ for } j \in \{2, \cdots, d\}, \end{cases}$$

where $\sim$ indicates that the probabilities hold upto $\mathcal{O}\left(\frac{1}{d^2}\right)$. This has a diagonal covariance matrix $\Lambda$ such that $\Lambda_{11} \sim 1 - \frac{d-1}{2d}$ and $\Lambda_{jj} \sim \frac{1}{2d}$ for $j \geq 2$, and convergence rate of $\frac{\text{Tr}\left(\Lambda^{-1}\Sigma\right)}{m} \sim \frac{1}{m}\left(\frac{2d}{d+1} \cdot \left(1 - \frac{d-1}{d^2}\right) + (d-1) \cdot 2d \cdot \frac{1}{d^2}\right) \leq \frac{4}{m}$, which does not grow with $d$!

**Logistic Regression.** Our second example is logistic regression for binary classification. In this case, $x \in \mathbb{R}^d$, $Y \in \{-1, 1\}$ and the negative log-likelihood function is: $L(y|x, \theta) = \log(1 + e^{-y\theta^\top x})$, and the corresponding Fisher information $I(x, \theta)$ is given as: $I(x, \theta) = \frac{e^{\theta^\top x}}{(1 + e^{\theta^\top x})^2} \cdot xx^\top$.

For illustration, suppose $\|\theta^*\|_2$ and $\|x\|_2$ are bounded by a constant and the covariance matrix $\Sigma$ is sandwiched between two multiples of identity in the PSD ordering i.e., $\frac{c}{d}I \preceq \Sigma \preceq \frac{C}{d}I$ for some constants $c$ and $C$. Then the regularity assumptions 1 and 2 are satisfied for constant values of $L_1, L_2, L_3$ and $L_4$. In this case, Theorem 1 states that choosing $m_1$ to be $\omega\left(\text{Tr}\left(I_U(\theta^*)^{-1}\right)\right) = \omega(d)$ gives us the optimal convergence rate of $(1 + o(1))\frac{\text{Tr}\left(I_{\Gamma^*}(\theta^*)^{-1}I_U(\theta^*)\right)}{m_2}$.

**Multinomial Logistic Regression.** Our third example is multinomial logistic regression for multiclass classification. In this case, $Y \in 1, \ldots, K$, $x \in \mathbb{R}^d$, and the parameter matrix $\theta \in \mathbb{R}^{(K-1) \times d}$. The negative log-likelihood function is written as: $L(y|x, \theta) = -\theta_y^\top x + \log(1 + \sum_{k=1}^{K-1} e^{\theta_k^\top x})$, if $y \neq K$, and $L(y = k|x, \theta) = \log(1 + \sum_{k=1}^{K-1} e^{\theta_k^\top x})$ otherwise. The corresponding Fisher information matrix is a $(K-1)d \times (K-1)d$ matrix, which is obtained as follows. Let $F$ be the $(K-1) \times (K-1)$ matrix with:

$$F_{ii} = \frac{e^{\theta_i^\top x}(1 + \sum_{k \neq i} e^{\theta_k^\top x})}{(1 + \sum_k e^{\theta_k^\top x})^2}, \quad F_{ij} = -\frac{e^{\theta_i^\top x + \theta_j^\top x}}{(1 + \sum_k e^{\theta_k^\top x})^2}$$

Then, $I(x, \theta) = F \otimes xx^\top$.

Similar to the example in the logistic regression case, suppose $\left\|\theta_y^*\right\|_2$ and $\|x\|_2$ are bounded by a constant and the covariance matrix $\Sigma$ satisfies $\frac{c}{d}I \preceq \Sigma \preceq \frac{C}{d}I$ for some constants $c$ and $C$. Since $F^* = \text{diag}\left(p_i^*\right) - p^*p^{*\top}$, where $p_i^* = P(y = i|x, \theta^*)$, the boundedness of $\left\|\theta_y^*\right\|_2$ and $\|x\|_2$ implies that $\widetilde{c}I \preceq F^* \preceq \widetilde{C}I$ for some constants $\widetilde{c}$ and $\widetilde{C}$ (depending on $K$). This means that $\frac{c\widetilde{c}}{d}I \preceq I(x, \theta^*) \preceq \frac{C\widetilde{C}}{d}I$ and so the regularity assumptions 1 and 2 are satisfied with $L_1, L_2, L_3$ and $L_4$ being constants. Theorem 1 again tells us that using $\omega(d)$ samples in the first step gives us the optimal convergence rate of maximum likelihood error.

# 6 Conclusion

In this paper, we provide an active learning algorithm for maximum likelihood estimation which provably achieves the optimal convergence rate (upto lower order terms) and uses only two rounds of interaction. Our algorithm applies in a very general setting, which includes Generalized Linear Models.

There are several avenues of future work. Our algorithm involves solving an SDP which is computationally expensive; an open question is whether there is a more efficient, perhaps greedy, algorithm that achieves the same rate. A second open question is whether it is possible to remove the *with replacement* sampling assumption. A final question is what happens if $I_U(\theta^*)$ has a high condition number. In this case, our algorithm will require a large number of samples in the first stage; an open question is whether we can use a more sophisticated procedure in the first stage to reduce the label requirement.

**Acknowledgements.** KC thanks NSF under IIS 1162581 for research support.

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
