[Supplementary Material]

# A Proofs

In order to prove Lemma 1, we use the following result which is a modification of [11]. In particular, the following lemma is a generalization of Theorem 5.1 from [11], and its proof (omitted here) follows from generalizing the proof of that theorem.

**Lemma 4.** *Suppose $\psi_1, \cdots, \psi_n : \mathbb{R}^d \to \mathbb{R}$ are random functions drawn iid from a distribution. Let $P = \mathbb{E}[\psi_i]$ and $Q : \mathbb{R}^d \to \mathbb{R}$ be another function. Let*

$$\widehat{\theta} = argmin_{\theta \in \mathcal{S}} \sum_i \psi_i(\theta), \text{ and } \theta^* = argmin_{\theta \in \mathcal{S}} P(\theta).$$

*Assume:*

1. *(Convexity of $\psi$): Assume that $\psi$ is convex (with probability one),*

2. *(Smoothness of $\psi$): Assume that $\psi$ is smooth in the following sense: the first, second and third derivatives exist at all interior points of $\mathcal{S}$ (with probability one),*

3. *(Regularity conditions): Suppose*

   (a) *$\mathcal{S}$ is compact,*
   (b) *$\theta^*$ is an interior point of $\mathcal{S}$,*
   (c) *$\nabla^2 P(\theta^*)$ is positive definite (and hence invertible),*
   (d) *$\nabla Q(\theta^*) = 0$,*
   (e) *There exists a neighborhood $B$ of $\theta^*$ and a constant $\widetilde{L_3}$ such that (with probability one), $\nabla^2 \psi(\theta)$ and $\nabla^2 Q(\theta)$ are $\widetilde{L_3}$ Lipschitz, namely*

   $$\left\| \left(\nabla^2 P(\theta^*)\right)^{-1/2} \left(\nabla^2 \psi(\theta) - \nabla^2 \psi(\theta')\right) \left(\nabla^2 P(\theta^*)\right)^{-1/2} \right\|_2 \leq \widetilde{L_3} \left\| \theta - \theta' \right\|_{\nabla^2 P(\theta^*)}, \text{ and}$$

   $$\left\| \left(\nabla^2 Q(\theta^*)\right)^{-1/2} \left(\nabla^2 Q(\theta) - \nabla^2 Q(\theta')\right) \left(\nabla^2 Q(\theta^*)\right)^{-1/2} \right\|_2 \leq \widetilde{L_3} \left\| \theta - \theta' \right\|_{\nabla^2 P(\theta^*)},$$

   *for $\theta, \theta' \in B$,*

4. *(Concentration at $\theta^*$) Suppose $\|\nabla \psi(\theta^*)\|_{\nabla^2 P(\theta^*)^{-1}} \leq \widetilde{L_1}$ and*

   $$\left\| \left(\nabla^2 P(\theta^*)\right)^{-1/2} \nabla^2 \psi(\theta^*) \left(\nabla^2 P(\theta^*)\right)^{-1/2} \right\|_2 \leq \widetilde{L_2}$$

   *hold with probability one.*

*Choose $p \geq 2$ and define*

$$\epsilon_n \stackrel{def}{=} \widetilde{c}(\widetilde{L_1}\widetilde{L_3} + \sqrt{\widetilde{L_2}})\sqrt{\frac{p \log dn}{n}},$$

*where $\widetilde{c}$ is an appropriately chosen constant. Let $\widetilde{c'}$ be another appropriately chosen constant. If $n$ is large enough so that $\sqrt{\frac{p \log dn}{n}} \leq \widetilde{c'} \min\left\{ \frac{1}{\sqrt{\widetilde{L_2}}}, \frac{1}{\widetilde{L_1}\widetilde{L_3}}, \frac{diameter(B)}{\widetilde{L_1}} \right\}$, then:*

$$(1 - \epsilon_n)\frac{\tau^2}{n} - \frac{\widetilde{L_1}^2}{n^{p/2}} \leq \mathbb{E}\left[Q(\widehat{\theta}) - Q(\theta^*)\right] \leq (1 + \epsilon_n)\frac{\tau^2}{n} + \frac{\max_{\theta \in \mathcal{S}} Q(\theta) - Q(\theta^*)}{n^p},$$

*where*

$$\tau^2 \stackrel{def}{=} \frac{1}{n^2} Tr\left( \left(\sum_{i,j} \mathbb{E}\left[\nabla \psi_i(\theta^*) \nabla \psi_j(\theta^*)^\top\right]\right) P(\theta^*)^{-1} Q(\theta^*) P(\theta^*)^{-1} \right).$$

The following lemma is a fundamental result relating the variance of the gradient of the log likelihood to Fisher information matrix for a large class of probability distributions [17].

**Lemma 5.** *Suppose $L$ satisfies the regularity conditions in Assumptions 1 and 2. Then, for any example $x$, we have:*

$$\mathbb{E}_{p(y|x,\theta^*)}\left[\nabla L(Y|x,\theta^*)\nabla L(Y|x,\theta^*)^\top\right] = \nabla^2 I_x(\theta^*).$$

We now prove Lemma 1.

*(Proof of Lemma 1).* We first define

$$\psi_i(\theta) = L\left(Y|X,\theta\right),$$

where $X \sim \Gamma$ and $Y \sim p(Y|X,\theta^*)$ for $i = 1,\cdots,m_2$ and $Q(\theta) \overset{\text{def}}{=} L_U(\theta)$. Using the notation of Lemma 4, this means that

$$\nabla^2 P(\theta^*) = I_\Gamma(\theta^*) \text{ and } \nabla^2 Q(\theta^*) = I_U(\theta^*).$$

Using the regularity conditions from Section 4 and the hypothesis that $I_\Gamma(\theta^*) \succeq cI_U(\theta^*)$, we see that this satisfies the hypothesis of Lemma 4 with constants

$$(\widetilde{L_1},\widetilde{L_2},\widetilde{L_3}) = (L_1/\sqrt{c}, L_2/c, L_3/c^{3/2})$$

We now apply Lemma 4 to conclude that for large enough $m_2$, we have:

$$(1-\epsilon_{m_2})\tau^2/m_2 - \frac{L_1^2}{cm_2^{p/2}} \leq \mathbb{E}\left[L_U\left(\widehat{\theta}\right) - L_U\left(\theta^*\right)\right] \leq (1+\epsilon_{m_2})\tau^2/m_2 + \frac{R}{m_2^p},$$

where

$$\epsilon_{m_2} = \mathcal{O}\left(\left(\widetilde{L_1}\widetilde{L_3} + \sqrt{\widetilde{L_2}}\right)\sqrt{\frac{p\log dm_2}{m_2}}\right) = \mathcal{O}\left(\frac{1}{c^2}\left(L_1 L_3 + \sqrt{L_2}\right)\sqrt{\frac{p\log dm_2}{m_2}}\right) \text{ and}$$

$$\tau^2 \overset{\text{def}}{=} \mathrm{Tr}\left(\mathbb{E}\left[\nabla\widehat{P}(\theta^*)\nabla\widehat{P}(\theta^*)^\top\right]I_\Gamma(\theta^*)^{-1}I_U(\theta^*)I_\Gamma(\theta^*)^{-1}\right) = \mathrm{Tr}\left(I_\Gamma(\theta^*)^{-1}I_U(\theta^*)\right),$$

using Lemma 5 in the last step. $\qquad\square$

We now prove Lemma 2.

*(Proof of Lemma 2).* Define

$$\psi_i(\theta) \overset{\text{def}}{=} L\left(Y|X,\theta\right),$$

where $X \sim U$ and $Y \sim p(Y|X,\theta^*)$ for $i = 1,\cdots,m_1$ and $Q(\theta) \overset{\text{def}}{=} \|\theta - \theta^*\|_2^2$. Using the regularity conditions from Section 4, we see that this satisfies the hypothesis of Lemma 4 with constants

$$(\widetilde{L_1},\widetilde{L_2},\widetilde{L_3}) = \left(L_1, L_2, \max\left(L_3, \frac{1}{\sqrt{\sigma_{\min}}}\right)\right))$$

We now apply Lemma 4 to conclude that

$$\mathbb{E}\left[\|\theta_1 - \theta^*\|_2^2\right] \leq (1+\epsilon_{m_1})\tau^2/m_1 + \frac{\text{diameter}(\Theta)}{m_1^2},$$

where $\epsilon_{m_1} = \mathcal{O}\left(\left(L_1\max\left(L_3, \frac{1}{\sqrt{\sigma_{\min}}}\right) + \sqrt{L_2}\right)\sqrt{\frac{\log dm_1}{m_1}}\right)$, and

$$\tau^2 \overset{\text{def}}{=} \mathrm{Tr}\left(\mathbb{E}\left[\nabla\widehat{L}_U(\theta^*)\nabla\widehat{L}_U(\theta^*)^\top\right]I_U(\theta^*)^{-2}\right) = \mathrm{Tr}\left(I_U(\theta^*)^{-1}\right),$$

using Lemma 5 in the last step. By the choice of $m_1$, we have that

$$\mathbb{E}\left[\|\theta_1 - \theta^*\|_2^2\right] \leq 2\tau^2/m_1.$$

Markov's inequality then tells us that with probability at least $1 - \delta$, we have:

$$\|\theta_1 - \theta^*\|_2^2 \leq \frac{2\tau^2}{\delta m_1} \leq \frac{1}{\beta^2 L_4^2}.$$

Using Assumption 2 on point-wise self concordance of $I(x,\theta)$ now finishes the proof. $\qquad\square$

*(Proof of Theorem 1).* The proof is a careful combination of Lemmas 1, 2 and 3.

**Lower Bound**: For any $\Gamma$ that satisfies $I_\Gamma(\theta^*) \succeq cI_U(\theta^*)$, we can apply Lemma 1 to write:

$$\mathbb{E}\left[L_U\left(\widehat{\theta}_\Gamma\right) - L_U\left(\theta^*\right)\right] \geq (1 - \epsilon_{m_2}) \frac{\text{Tr}\left(I_\Gamma(\theta^*)^{-1}I_U(\theta^*)\right)}{m_2} - \frac{L_1^2}{cm_2^2}.$$

The lower bound follows.

**Upper Bound**: We begin by showing that if Assumptions 1 and 2 are satisfied, then, from Lemma 2, we have that with probability $\geq 1 - \delta$, it holds that:

$$\frac{\beta - 1}{\beta} I(x, \theta^*) \preceq I(x, \theta_1) \preceq \frac{\beta + 1}{\beta} I(x, \theta^*) \ \forall \ x \in U$$

with probability $\geq 1 - \delta$. This means that the following hold for distributions $\Gamma_1$, $\Gamma^*$ and $U$:

$$\frac{\beta - 1}{\beta} I_{\Gamma_1}(\theta^*) \preceq I_{\Gamma_1}(\theta_1) \preceq \frac{\beta + 1}{\beta} I_{\Gamma_1}(\theta^*), \tag{5}$$

$$\frac{\beta - 1}{\beta} I_{\Gamma^*}(\theta^*) \preceq I_{\Gamma^*}(\theta_1) \preceq \frac{\beta + 1}{\beta} I_{\Gamma^*}(\theta^*), \text{ and} \tag{6}$$

$$\frac{\beta - 1}{\beta} I_U(\theta^*) \preceq I_U(\theta_1) \preceq \frac{\beta + 1}{\beta} I_U(\theta^*). \tag{7}$$

Since $\overline{\Gamma} = \alpha\Gamma_1 + (1-\alpha)U$, we have that $I_{\overline{\Gamma}}(\theta^*) \succeq \alpha I_{\Gamma_1}(\theta^*)$ which further implies that $I_{\overline{\Gamma}}(\theta^*)^{-1} \preceq \frac{1}{\alpha}I_{\Gamma_1}(\theta^*)^{-1}$. Similarly, since $I_{\overline{\Gamma}}(\theta^*) \succeq (1 - \alpha)I_U(\theta^*)$, we can apply Lemma 1 on $\overline{\Gamma}$ to get:

$$\mathbb{E}\left[L_U\left(\theta_2\right) - L_U\left(\theta^*\right)\right] \leq (1 + \widehat{\epsilon}_{m_2}) \frac{\text{Tr}\left(I_{\overline{\Gamma}}(\theta^*)^{-1}I_U(\theta^*)\right)}{m_2} + \frac{R}{m_2^2} \leq \frac{1}{\alpha}(1 + \widehat{\epsilon}_{m_2}) \frac{\text{Tr}\left(I_{\Gamma_1}(\theta^*)^{-1}I_U(\theta^*)\right)}{m_2} + \frac{R}{m_2^2}$$

$$\leq (1 + \widetilde{\epsilon}_{m_2}) \frac{\text{Tr}\left(I_{\Gamma_1}(\theta^*)^{-1}I_U(\theta^*)\right)}{m_2} + \frac{R}{m_2^2},$$

where $\widehat{\epsilon}_{m_2}, \widetilde{\epsilon}_{m_2} = \mathcal{O}\left(\frac{1}{(1-\alpha)^2}\left(L_1L_3 + \sqrt{L_2}\right)\sqrt{\frac{\log dm_2}{m_2}}\right) = \mathcal{O}\left(\left(L_1L_3 + \sqrt{L_2}\right)\frac{\sqrt{\log dm_2}}{m_2^{1/6}}\right)$.

From (5) and (7), the right hand side is at most:

$$(1 + \widetilde{\epsilon}_{m_2})(\frac{\beta + 1}{\beta - 1})^2 \frac{\text{Tr}\left(I_{\Gamma_1}(\theta_1)^{-1}I_U(\theta_1)\right)}{m_2} + \frac{R}{m_2^2}$$

By definition of $\Gamma_1$, this is at most:

$$(1 + \widetilde{\epsilon}_{m_2})(\frac{\beta + 1}{\beta - 1})^2 \frac{\text{Tr}\left(I_{\Gamma^*}(\theta_1)^{-1}I_U(\theta_1)\right)}{m_2} + \frac{R}{m_2^2}$$

Finally, applying (6) and (7), we get that this is at most:

$$(1 + \widetilde{\epsilon}_{m_2})(\frac{\beta + 1}{\beta - 1})^4 \frac{\text{Tr}\left(I_{\Gamma^*}(\theta^*)^{-1}I_U(\theta^*)\right)}{m_2} + \frac{R}{m_2^2}$$

The upper bound follows. $\qquad\square$