[Reviews · NeurIPS 2015]

Submitted by Assigned_Reviewer_1

This paper analyzes active learning for maximum likelihood estimation of a certain general class of models, and provides a theoretical bound on the number of samples that are needed to be labeled to achieve a given error on the expected log likelihood. As a consequence, active learning for MLE of such models requires only a single round of querying for labels. This is an interesting result. The main weakness of the current paper is the bound on m_1. That bound depends on \theta*, which is what we want to approximate by using m_1 samples. In principle, m_1 could be arbitrarily large and defeat the whole point of doing active learning. I was surprised that there was no mention of that anywhere, but just saw that it is only brought up in the very last sentence.

One minor thing: lien 4 in algorithm 1, should have \Gamma_1 = \frac{a*}{m_2} or be \gamma_i = \frac{a^*_i}{m_2}
Summary: Fine paper proving a theoretical bound on the number of samples needed for active learning for MLE. But the bound itself depends on an estimation step that could take a large number of samples.

Submitted by Assigned_Reviewer_2

This paper studies the problem of active learning for maximum likelihood estimation.

One considers a pool of examples and a space of models, and then attempts to query examples in order to learn the parameters of the model maximizing the loglikelihood of the sample's (unknown) labels.

A two-step algorithm is proposed, first sampling uniformly, then selecting queries based on solving an SDP problem (which may therefore not scale very well in practice?).

I'm a bit worried by the several assumptions made.

The most important one (condition 1) holds for generalized linear models, but it is not very clear to me how constraining this assumption is.

The explanation is reasonably clear, and the derivation seems correct.

As the authors say, a maximum likelihood result is interesting as most existing results focus on PAC settings.

Still, is this case so much different, and why?

It would have been more interesting and convincing if a comparison with PAC results would be provided which are applicable to a similar setting (except for the difference loglikelihood vs. classification).

Summary: The result is interesting and seems correct.

It would have been interesting to see a comparison with existing (PAC-based) results, and to have a better description of the limitations of the approach (e.g. due to the assumptions, not just referring to future work to relax them) and of the scalability.

Submitted by Assigned_Reviewer_3

The paper addresses the problem of active learning by trying to find the MLE from a parametrized class of models. It shows that under certain conditions an expression of the Fisher information is a suitable surrogate objective function. Optimizing this surrogate objective helps find a \emph{good} sampling strategy. However, this expression entails the optimal parameter \theta^* whose approximation is the ultimate goal of the algorithm. To overcome this issue the paper takes a two step approach. In the first step a fraction of the budgeted queries is used to get a rough estimate of \theta^* which is subsequently used to optimize the surrogate objective function and obtain a good sampling distribution. The paper shows that using this two step approach one can achieve an asymptotically optimal error given a fixed budget on number of queries for certain classes of models which include GLMs.

The quality and presentation of the paper is good. The result is significant, at least from a theoretical perspective, albeit being slightly disappointing since it shows that a two step batch sampling is asymptotically as good as one can get with a full fledged active learning on these classes of models. This aspect brings to mind the question of applicability of the result to practical settings and strongly begs some numerical demonstration and comparison which is unfortunately missing in the paper.

comments: 1. diameter(\Theta) in Theorem 1 is not defined. 2. The big O notation has in itself an implicit inequality and there is no proper definition of f > O(g) that I am aware of. Hence the author(s) should clarify the condition in Theorem 1 which states that m > O(...). 3. It would be informative to see what are the practical implication of Assumption 1 vis-ae-vis the result of Theorem 1. For example, if the set U was randomly chosen from \mathcal{X} with some reasonable distribution what is the scaling of \sigma_min in part 3 of Assumption 1 and what would be the implication for m_1. At its current form the paper does not give any idea under what regime this is an applicable method. The lack of any numerical demonstration even in the supplementary section is not helpful. While the theoretical contribution is certainly valuable demonstration of the applicability of the result makes it many folds more interesting. Furthermore, if it is possible to demonstrate the method in practical scenarios this type of result strongly begs some comparison with other methods.
Summary: Interesting solution to the problem of maximum likelihood estimation in the context of active learning. However, it needs some clarification and demonstration of the result.

Submitted by Assigned_Reviewer_4

The authors pay attentions to another active learning setting, i.e., maximum likelihood estimation, and propose an algorithm with two steps: query the labels of a number of random samples to construct an estimation of the optimal and then select a set of samples for label query by optimizing the estimation by minimizing the expected log-likelihood error. However, I have the following comments:

What does that the Fisher information matrix for any model at any (x, y) does not depend on y imply? I think it implies that the learning problem should be easy to solve, e.g., it can be solved by linear models. The examples that the authors provide are also linear models. General active learning algorithm should solve more complex problems. Furthermore, this condition states that there is no y in the Fisher matrix, so the computation is much simpler. The result in this paper is based on too many strong assumptions, i.e., Condition 1, Assumption 1, and Assumption 2. Assumption 1 includes 6 more conditions. What real applications can satisfy all these conditions? Why are all these conditions necessary? Just for simple computation? Furthermore, the paper only focuses on the trusductive setting. I do not understand why the authors need to define what the optimal is (Definition 1). What is the value of m_{2} in Lemma 1? m_{2} depends on p, the authors should provide its value clearly, e.g., m_{2}>=XXX.

What does the condition that I_{\Gamma}(\theta^{*})\succeq c I_{U}(\theta^{*}) mean in Lemma 1? I think it states that \Gamma is an approximation of U (with constant c). However, the quality of \Gamma should depend on m_{2}, so the condition that I_{\Gamma}(\theta^{*})\succeq c I_{U}(\theta^{*}) also depends on m_{2}. It is not reasonable to assumption that I_{\Gamma}(\theta^{*})\succeq c I_{U}(\theta^{*}) directly. In line 155, what is Tr(.)? Trace of a matrix? The author should make this clearly. In line 186, without steps 1-2, how to solve step 3 without \theta_{1}? This authors claim that "Notably, these works provide promising empirical evidence for this approach to active learning", why there is no experimental study?
Summary: In this paper, the authors try to provide an active learning algorithm for maximum likelihood estimation with many strong assumptions.

Author Feedback
Author rebuttal: We thank all reviewers for their feedback. We emphasize that prior to our work, theoretical results on AL were known only for binary classification and linear regression, and thus the main contribution of our work is to substantially broaden the scope of active learning theory.

***For ALL reviewers***

-- "Lack of adaptiveness" due to just one additional round: A major "take-home message" of our paper is that unlike the PAC case, only an extra round of adaptivity is sufficient to achieve optimal performance in MLE.

-- "How large is m_1 and when is it reasonable": m_1, the number of samples in the first stage, is a property of the problem, and does not depend on m_2. As suggested by reviewer 2, we will do some simple examples of distributions where m_1 is reasonable to give a feel of the result.

*** Reviewer 3 ***

-- A major concern in this review is that of Assumption 1, which we feel is not justified. This is an entirely standard set of assumptions in theoretical statistics, under which the convergence rates are studied (and proved) for maximum likelihood estimation problems - see [5, 21] for example. Since we are looking to obtain an optimal convergence rate, this is a very natural assumption for our setting.

- Condition 1 (Fisher information independent of y): While we agree that not all MLE problems obey this condition, it is obeyed by the class of Generalized Linear Models, which include **multiclass classification**, **linear regression**, and **conditional random fields**. Theoretical results on AL were previously known only for binary classification and linear regression, and thus our work substantially broadens the scope of AL theory.

- Lack of (numerical) demonstration: [11] and [12] have already conducted extensive experiments on real data on a multi-round version of our algorithm to obtain promising results. We have chosen not to repeat them as they don't add value to this paper.

- "Defining what the optimal is (Definition 1)": This notion of optimal sampling distribution is one of the main contributions of our work. Theorem 1 tells us why this notion of optimality is important: under regularity conditions, it achieves the best possible error since the upper bound (Eqn (1)) matches the lower bound (Eqn(2)).

- "Value of m_2 in Lemma 1": The bound on m_2 comes by setting \epsilon_{m_2} < 1. We will clarify this in the final version.

- "What does the condition I_{\Gamma}(\theta^*).... mean": In the statement of Lemma 1, \Gamma is a fixed distribution; I_{\Gamma}(\theta^*) is a property of this distribution, and both of these have no dependence on m_2. (Only \hat{\theta}_{\Gamma} depends on m_2.)

*** Reviewer 5 ***

We thank Reviewer 5 for the suggestion of comparing with PAC; but we note that the PAC and MLE settings are very different, and it is unclear how to do a direct yet fair comparison. The gains due to active learning are more modest for PAC learning with noise (under the agnostic or Tsybakov noise conditions). An open question is whether a limited number of rounds of adaptivity are sufficient for PAC learning in these contexts.